genetics/plant science

*Actinidia chinensis*, colchicine, tetraploid, identification, real-time quantitative PCR, transcriptome analysis

**Author for correspondence:**
Hanyao Zhang
e-mail: zhanghanyao@hotmail.com

# Induction, identification and genetics analysis of tetraploid *Actinidia chinensis*

Shengxing Li[1], Xiaozhen Liu[1], Huiming Liu[1], Xianang Zhang[2], Qinxia Ye[2] and Hanyao Zhang[1,2]

[1]Key Laboratory for Forest Resources Conservation and Utilization in the Southwest Mountains of China, and [2]Key Laboratory of Biodiversity Conservation in Southwest China, State Forest Administration, Southwest Forestry University, Kunming, Yunnan Province, 650224, People's Republic of China

HZ, 0000-0001-7440-4348

*Actinidia chinensis* is a commercially important fruit, and tetraploid breeding of *A. chinensis* is of great significance for economic benefit. In order to obtain elite tetraploid cultivars, tetraploid plants were induced by colchicine treatment with leaves of diploid *A. chinensis* 'SWFU03'. The results showed that the best treatment was dipping leaves 30 h in 60 mg l$^{-1}$ colchicine solutions, with induction rate reaching 26%. Four methods, including external morphology comparison, stomatal guard cell observation, chromosome number observation and flow cytometry analysis were used to identify the tetraploid of *A. chinensis*. Using the induction system and flow cytometry analysis methods, 187 tetraploid plants were identified. Three randomly selected tetraploid plants and their starting diploid plants were further subjected to transcriptome analysis, real-time quantitative polymerase chain reaction (RT-qPCR) and methylation-sensitive amplification polymorphism (MSAP) analysis. The transcriptome analysis results showed that there were a total of 2230 differentially expressed genes (DEG) between the diploid and tetraploid plants, of which 660 were downregulated and 1570 upregulated. The DEGs were mainly the genes involved in growth and development, stress resistance and antibacterial ability in plants. RT-qPCR results showed that the gene expression levels of the growth and stress resistance of tetraploid plants were higher than those of diploid ones at the transcriptome level. MSAP analysis of DNA methylation results showed that tetraploid plants had lower methylation ratio than diploid ones. The present results were valuable to further explore the epigenetics of diploid and tetraploid kiwifruit plants.

# 1. Introduction

*Actinidia chinensis* is a commercially important fruit, and its chromosomes are small, but a lot [1]. It seems to have a lot of ploidy level variation [2,3], including from diploid to octoploid and there may be chromosome races within individual taxa [4]. The basic chromosome number of *A. chinensis* is 29, and most of the *A. chinensis* plants are diploid ($2n = 2x = 58$). It is hard to find tetraploid, hexaploid and octoploid plants, yet, they do exist in nature [1,4,5]. Differences in ploidy level have made hybridization difficult due to infertility of polyploidy; however, it can offer significant opportunities for creating novel types [6]. Tetraploid fruit trees usually have better stress resistance, bigger fruit and faster growth speed compared to diploid ones [6–8], hence induction and identification of tetraploids are very important for fruit tree breeding. Researchers had attempted ploidy level manipulation on *A. chinensis* [6–8].

Treatment of plant tissues with colchicine is the most commonly used method for inducing tetraploidy [9,10]. In practice, colchicine induction of tetraploidy was commonly used with immersion or dropping method [11,12]. *In vitro* induction method has a lot of advantages, not only experimental conditions can be artificially controlled, easy operation, low cost, but also the results of the experiment can be replicated [13].

Current common methods for ploidy identification were chromosome counting [14,15] and flow cytometry testing [16,17]; external morphology observation [18,19] and stomatal guard cells microscopic observation were auxiliary appraisals. Although these methods were used widely, each method has its disadvantages. For example, external morphology observation and stomatal guard cell microscope observation were simple but not accurate enough; chromosome counting was accurate but time-consuming and difficult to conduct; flow cytometry testing was fast and high flux, but just relatively accurate [20].

Transcriptome sequencing (RNA-seq) is based on the whole genome of high-throughput sequencing technologies [21,22]. RNA-seq is a modern tool of transcriptome analysis. A new generation of sequencing technology can be used for fast and accurate transcriptome profiling [23]. This technology can detect any species of the overall transcription activity; based on the analysis of the transcriptome, gene structure and expression level can be described. This method can also detect novel transcript and rare transcript, providing an experimental study more comprehensive set of transcription information. The results of RNA-seq are highly reproducible, having made RNA-seq an increasingly attractive and popular method for studying transcriptomes [24–27].

Real-time fluorescent quantitative polymerase chain reaction (RT-qPCR) refers to adding fluorescent groups in the PCR reaction system, using the fluorescent signal cumulative to monitor the entire process of PCR. It achieves the leap from qualitativeness to quantitativeness, with strong specificity, high sensitivity, good repeat-ability, accuracy and quickness, having become an important tool in molecular biology research [28,29]. The expression levels of specific genes in transcriptome of some species were successfully confirmed by RT-qPCR [30–32].

DNA methylation has been reported as one of the most common covalent modifications of DNA, and has both epigenetic and mutagenic effects causing specific gene expression, cell differentiation, chromatin inactivation and embryo growth [33,34]. Methylation-sensitive amplification polymorphism (MSAP) combined with the appealing features of amplified fragment length polymorphism (AFLP) is based on the restriction enzyme and PCR amplification, and is highly efficient for large-scale detection of cytosine methylation in the genome [35,36].

Colchicine-induced chromosome-doubling tends to increase stress resistance of plants [37,38]. However, there is no evidence to show if the resistance of *A. chinensis* would increase the gene expression level after chromosome-doubling induction using colchicine or not. To lay the foundation of *A. chinensis* tetraploid breeding and understand what has changed on the gene expression levels in the *A. chinensis* tetraploid, it is necessary to carry out research on the induction, identification and genetics analysis of *A. chinensis* tetraploid plants. Here, we present the results of using young plants of *A. chinensis* 'SWFU03' as materials for inducing tetraploid plants and employing four methods including external morphology comparison, stomatal guard cell observation, chromosome number observation and flow cytometry analysis to identify the tetraploid. Then transcriptome analysis, fluorescence quantitative PCR and MSAP analysis were further employed to access the diploid and tetraploid plants.

# 2. Material and methods

## 2.1. Plant materials

Young plants of 'SWFU03' were used as the materials, and leaf blades were used as explants. Leaves were taken from Key Laboratory of Biodiversity Conservation in Southwest China, Southwest Forestry

University. 'SWFU03' is an excellent *A. chinensis* breeding clone selected by Southwest Forestry University. Donor diploid plants was used as references for phenotypic and genetic analysis of resulted tetraploids, and the culture room conditions were as following: the temperature was $25 \pm 1°C$, humidity was 40–80%, the light intensity was 2000 LX, and lighting time 16 h light/8 h darkness, with the incandescent light source.

## 2.2. Colchicine treatment

For initiation of *in vitro* culture, leaves were sliced into $1 \times 1$ cm pieces on the clean bench, and then were put into sterilized colchicine solution. The six colchicine concentrations were 0, 20, 40, 60, 80 and $100 \text{ mg l}^{-1}$, and treatment times were 6, 12, 18, 24 and 30 h, each treatment time had a control with sterile water treatment. The orthogonal experimental design was employed in the study. Every group had 20 explants, 30 groups in total and three repeats. In total, 500 ml glass bottles with plastic cap were used as culture vessels; every glass bottle had four explants. Using MS medium [39] + 0.5 mg l$^{-1}$ NAA (1-naphthy-ceticacid) + 5 mg l$^{-1}$ 6-BA (6-benzyladenine) + sucrose 30 g l$^{-1}$ + agar 5 g l$^{-1}$, pH 5.8, autoclaved at 121°C for 20 min before use. Randomized block was used to arrange the culture vessels on the culture shelf, after cultured for 30 d, and total contamination rate [(total contamination/total explant) × 100%], dieback rate [(total dieback/total explant) × 100%] and polyploidization efficiency [(number of polyploidizations/explant of survival number) × 100%] were determined.

For subculturing, regeneration and shoot propagation, the medium for regeneration was the same as initiation culture. The medium for shoot propagation was ½ MS + 0.7 mg l$^{-1}$ IBA (3-Indolebutyric acid) + 0.1 g l$^{-1}$ AC (activated carbon) + sucrose 30 g l$^{-1}$ + agar 5.5 g l$^{-1}$.

## 2.3. Phenotypic evaluation

Diploid and tetraploid *A. chinensis* plants with the same successive generation were randomly selected. In 10 tetraploid and 10 diploid plants, two leaves were collected from each plant, and a total of 40 leaves were measured. The third and fourth leaves from each plant after subculturing for rooting for one month were sampled. Twenty centimetres of straight edge was used to measure the leaf length and width, respectively, and leaf thickness was measured with a vernier caliper. Finally, the average value was generated and compared.

Fifteen diploid and 15 tetraploid *A. chinensis* leaves were randomly selected from the same successive generation plants. The differences of stomatal density, length, and width of guard cell between diploid and tetraploid leaves were observed using a Leica DMR-X microscope (Leica, Stuttgart, Germany) at 40×. Data were analysed using SPSS 17. Descriptive statistics, one-way ANOVA test, and independent *t*-test were used [40].

The root tips of 10 diploid and 10 tetraploid *A. chinensis* plants *in vitro* culture were collected, and put into 0.002 mol l$^{-1}$ 8-hydroxyquinoline solution for 1–3 h, then stored at 4°C for 16 h. Subsequently, they were fixed using a solution of alcohol : glacial acetic acid = 3 : 1 for 12–24 h, hydrolysed in 15% HCl solution for 3–5 min. Meanwhile, carbonic acid magenta was used for staining 15–25 min and the chromosome number observed using an optical microscope after the roots were placed on the slides. Finally, photos were taken using a camera for counting chromosome number [41,42]. Five cells were used for chromosome counting in each sample.

## 2.4. Flow cytometry analysis

Fresh leaves of three diploid and 187 *A. chinensis* plants which were suspected to be tetraploid according to the morphological indexes were used as the material for flow cytometry analysis of the relative nuclear DNA content according to a previous report [43]. Samples were calibrated against standards of diploid and tetraploid *A. chinensis* for which the ploidy had been checked previously by morphological indexes like thicker leaves, more leaf pubescence, bigger stems and larger veins. Using *Oryza sativa* as internal standard, and Germany Partec Flow cytometry (CyFlow-Space-3000) for detection, at least 5000 nuclei were collected in each measurement and repeated at least three times for each of the species.

## 2.5. Transcriptome analysis

To compare the differential expression of genes (DEGs) in three diploid and three tetraploid plants, the fresh leaves were used as the material and next-generation sequencing was employed to access the RNA samples. RNA-seq was performed by Nextomics Biosciences Ltd using Illumina HiSeqTM 2500.

**Table 1.** Primer sequence of target genes (TG) and reference genes (RG) used in RT-qPCR.

| gene name | primer | sequence (5′–3′) | length (bp) |
|---|---|---|---|
| elongation factor EC 3.6.5.3 (RG) | forward primer | ACAAGCTGGTGACAATGTGG | 127 |
| | reverse primer | CGACCACCTTCATCCTTTGT | |
| β-1,3-glucosidase (TG) | forward primer | ATGCTCGTGACAGGAAACGC | 109 |
| | reverse primer | GCAATGCCAATGTAACACCTGC | |
| triacylglycerol lipase (TG) | forward primer | ATGCTCGTGACAGGAAACGC | 124 |
| | reverse primer | GCAATGCCAATGTAACACCTGC | |
| Achn295071 (TG) | forward primer | CGTCGAAGCAGGGTCATTTA | 96 |
| | reverse primer | TTGAGCCTCTGGATTGGTAAAG | |
| Achn247401 (TG) | forward primer | CTGTCCGGAATAACCCTAACC | 113 |
| | reverse primer | GTGACCAGGAACATGACTATCC | |
| Achn140601 (TG) | forward primer | CGGTGTTCTCGTGGATGTATAG | 117 |
| | reverse primer | CCGCCTTGTCCTTCATCTC | |
| Achn047951(TG) | forward primer | CTTGGAAGGCTCGCTCTATG | 103 |
| | reverse primer | TCCCTCTGGGAACACAAATG | |
| Achn048611(TG) | forward primer | GTATTGGAAGGCACGCTCTA | 105 |
| | reverse primer | TCCCTCTGGGAACACAAATG | |
| Achn217451(TG) | forward primer | TGAATGTTGGAGACGGAAAGG | 109 |
| | reverse primer | TCAGGAACTGATGGTGTTGAAG | |
| Achn125011(TG) | forward primer | TGGGAGGTCACCTACATACTT | 94 |
| | reverse primer | CATCGCAAACTCCCAACATTC | |
| Achn009011(TG) | forward primer | GACACAAAGCTAGGGACGATAC | 111 |
| | reverse primer | CTCTCCTTGCTTTAGGCTCTTC | |
| Achn007531(TG) | forward primer | CTACGGTGTCGTGATCCTAGA | 114 |
| | reverse primer | GTCTTCACTGCCCTCACAAA | |

FastQC (http://www.bioinformatics.babraham.ac.uk/projects/fastqc/) software was used for clean data quality control. The Kiwifruit reference sequence was downloaded from http://bioinfo.bti.cornell.edu/cgi-bin/kiwi/download.cgi, and transcriptome analysis was performed using DEGseq and DESeq softwares [44]. Blast2GO [45] was applied to obtain GO annotations of unigenes; after the GO annotation of each unigene was obtained, WEGO software [46] was used to classify and count the GO functions of all unigenes. Pathway annotations of the unigenes based on the KEGG database were also obtained.

## 2.6. RT-qPCR analysis

Tissue culture shoots leaves of three diploid and three tetraploid *A. chinensis* plants were used as the materials. The total RNA was extracted using the Qiagen RNeasy Mini Kit (Qiagen Inc., Valencia, CA), and then the RNA was reverse transcribed into cDNA by random primers. The RT-qPCR was conducted according to a previous report [47].

The transcriptional expression level of 11 candidate genes including two known resistance genes which were β-1,3-glucosidase [48] and triacylglycerol lipase [49] were checked by RT-qPCR in diploid and tetraploid kiwifruit plants. Gene specific primers were designed using Primer Premier 5.0 software, and the primers used for RT-qPCR analyses are listed in table 1. The $2^{(-\Delta\Delta Ct)}$ method [47] was employed to analyse the data. Statistical significance was evaluated by the two-sample *t*-test (independent variable) at the 95% confidence level.

**Table 2.** Adapter and primer sequences.

| adapter and primer | sequences |
| --- | --- |
| EcoRIAdapter 1 | CTCGTAGACTGCGTACC |
| EcoRIAdapter 2 | AATTGGTACGCAGTC |
| HM Adapter 1 | GACGATGAGTCTAGAA |
| HM Adapter 2 | CGTTCTAGACTCATC |
| PreAmp primer E-A | GACTGCGTACCAATTCA |
| PreAmp primer HM-T | GATGAGTCTAGAACGGT |
| selective primers E32 | GACTGCGTACCAATTCAAC |
| selective primers E45 | GACTGCGTACCAATTCATG |
| selective primers Msp39 | GATGAGTCCTGAGCGGAGA |
| selective primers Msp40 | GATGAGTCCTGAGCGGAGC |
| selective primers Msp41 | GATGAGTCCTGAGCGGAGG |
| selective primers Msp44 | GATGAGTCCTGAGCGGATC |
| selective primers Msp48 | GATGAGTCCTGAGCGGCAC |
| selective primers Msp50 | GATGAGTCCTGAGCGGCAT |
| selective primers Msp61 | GATGAGTCCTGAGCGGCTG |

## 2.7. MSAP analysis

Three deploid and three tetraploid plants which confirmed by chromosome number observation and flow cytometry analysis were subjected to analysed DNA methylation using MSAP. The MSAP system had three major parts, digestion and ligation reactions, preamplification and selective amplification reactions, and detection reactions. The designed adapters and primers for *Eco*R I and *Hpa* II-*Msp* I listed in table 2 were fixed as described by Lu *et al.* [50], with some modifications. We used 16 pairs of selective primers obtained from four *Eco*R I primers (E32 and E45) in combination with four *Hpa* II/*Msp* I primers (Msp39, Msp40, Msp41 Msp44, Msp48, Msp50 and Msp61), to analyse DNA methylation in CCGG sites of diploid and tetraploid tissues.

# 3. Results

## 3.1. Polyploidization efficiency

Multiple comparisons of the fatality rate and mutagenesis rate showed that the fatality rate and mutagenesis rate were significantly different among the treatment combinations. The treatment combination 7 ($12\,h + 20\,mg\,l^{-1}$) and 9 ($24\,h + 20\,mg\,l^{-1}$) had the lowest dieback rate, and the treatment combination 15 ($30\,h + 60\,mg\,l^{-1}$) had the highest polyploidization efficiency (table 3).

As shown in table 3 and figure 1, when colchicine concentration was $20\,mg\,l^{-1}$, the mutagenesis rate and dieback rate were the lowest. When colchicine concentration was greater than $20\,mg\,l^{-1}$, the change between dieback rate and colchicine concentration was not obvious. The polyploidization efficiency first increased and then declined with the increase of colchicine concentration, and reached the highest at $60\,mg\,l^{-1}$.

To sum up, polyploidization efficiency rose with increasing concentration of colchicine and gradually appeared to be stable to a certain degree. For increasing the polyploidization efficiency, a high concentration of colchicine and long processing time were required. Meanwhile, a low concentration of colchicine led to a low polyploidization efficiency, and was not suitable for induction of tetraploid *A. chinensis* plants.

When treatment time was brief, the polyploidization efficiency increased gradually with an increase of colchicine concentration. When treatment time was long, the polyploidization efficiency tended to decrease with an increase of colchicine concentration. However, with the same concentration, the treatment time did not significantly affect the polyploidization efficiency. These results showed that the main influence factor in the induction of tetraploid plants was the concentration of colchicine. Exploiting this system, a total of 187 independent tetraploid plants were generated.

**Table 3.** Polyploidization efficiency using *in vitro* colchicine treatment of leaf explants of *A. chinensis*. Mean separation within column by one-way ANOVA. Duncan's multiple comparison analysis was performed. The means ($n = 20$) followed by the same letter do not differ at $p = 0.05$.

| colchicine concentration (mg l$^{-1}$) | time (h) | contamination rate (%) | dieback rate (%) | polyploidization efficiency (%) |
|---|---|---|---|---|
| 0 | 6 | 20.0 | 0.0 | 0.00 |
|  | 12 | 20.0 | 0.0 | 0.00 |
|  | 18 | 20.0 | 0.0 | 0.00 |
|  | 24 | 20.0 | 0.0 | 0.00 |
|  | 30 | 20.0 | 0.0 | 0.00 |
| 20 | 6 | 0.0 | 10.0$^{D}$ | 0.00 |
|  | 12 | 0.0 | 5.0$^{D}$ | 0.0012 |
|  | 18 | 20.0 | 6.3$^{D}$ | 0.00 |
|  | 24 | 0.0 | 5.0$^{D}$ | 0.00 |
|  | 30 | 20.0 | 12.5$^{CD}$ | 0.00 |
| 40 | 6 | 20.0 | 50.2$^{A}$ | 0.00 |
|  | 12 | 20.0 | 12.5$^{CD}$ | 9.52$^{C}$ |
|  | 18 | 20.0 | 37.5$^{BC}$ | 10.00$^{BC}$ |
|  | 24 | 0.0 | 30.0$^{BC}$ | 7.14$^{C}$ |
|  | 30 | 40.0 | 16.7$^{CD}$ | 10.00$^{BC}$ |
| 60 | 6 | 0.0 | 38.3$^{BC}$ | 16.22$^{AB}$ |
|  | 12 | 0.0 | 43.3$^{AB}$ | 14.71$^{B}$ |
|  | 18 | 20.0 | 37.5$^{BC}$ | 20.00$^{A}$ |
|  | 24 | 40.0 | 33.3$^{BC}$ | 25.00$^{A}$ |
|  | 30 | 0.0 | 16.7$^{C}$ | 26.00$^{A}$ |
| 80 | 6 | 40.0 | 36.1$^{BC}$ | 13.04$^{BC}$ |
|  | 12 | 20.0 | 31.3$^{B}$ | 12.12$^{BC}$ |
|  | 18 | 0.0 | 20.0$^{C}$ | 10.42$^{BC}$ |
|  | 24 | 20.0 | 56.3$^{A}$ | 19.05$^{AB}$ |
|  | 30 | 0.0 | 20.0$^{C}$ | 14.58$^{B}$ |
| 100 | 6 | 20.0 | 43.8$^{AB}$ | 22.22$^{A}$ |
|  | 12 | 20.0 | 25.0$^{C}$ | 16.67$^{AB}$ |
|  | 18 | 20.0 | 37.5$^{B}$ | 20.00$^{A}$ |
|  | 24 | 20.0 | 35.4$^{BC}$ | 9.68$^{C}$ |
|  | 30 | 0.0 | 40.0$^{B}$ | 13.89$^{BC}$ |

## 3.2. Differences in the morphologies and stomatal guard cells between diploid and tetraploid plants

Table 4 shows differences between leaves of diploidy and tetraploidy. The average leaf length of diploid was 3.57 cm, and that of tetraploid was 3.72 cm and the difference was significant. The average leaf width of diploidy (2.26 cm) was not significantly from that of tetraploid (2.35 cm). The average leaf thickness of diploidy (0.52 mm) was significantly ($p = 0.004$) lower than that of tetraploid (1.04 mm). Tetraploid plants exhibited thicker leaves, more leaf pubescence, bigger stems and larger veins (figure 2a,b). According to morphological indexes like thicker leaves, more leaf pubescence and so on, 187 suspected tetraploid plants were selected.

There was a major difference between the stomatal characteristics of the leaves of the diploid and tetraploid plants. Stomatal size of tetraploid was obviously larger than that of diploid, and the

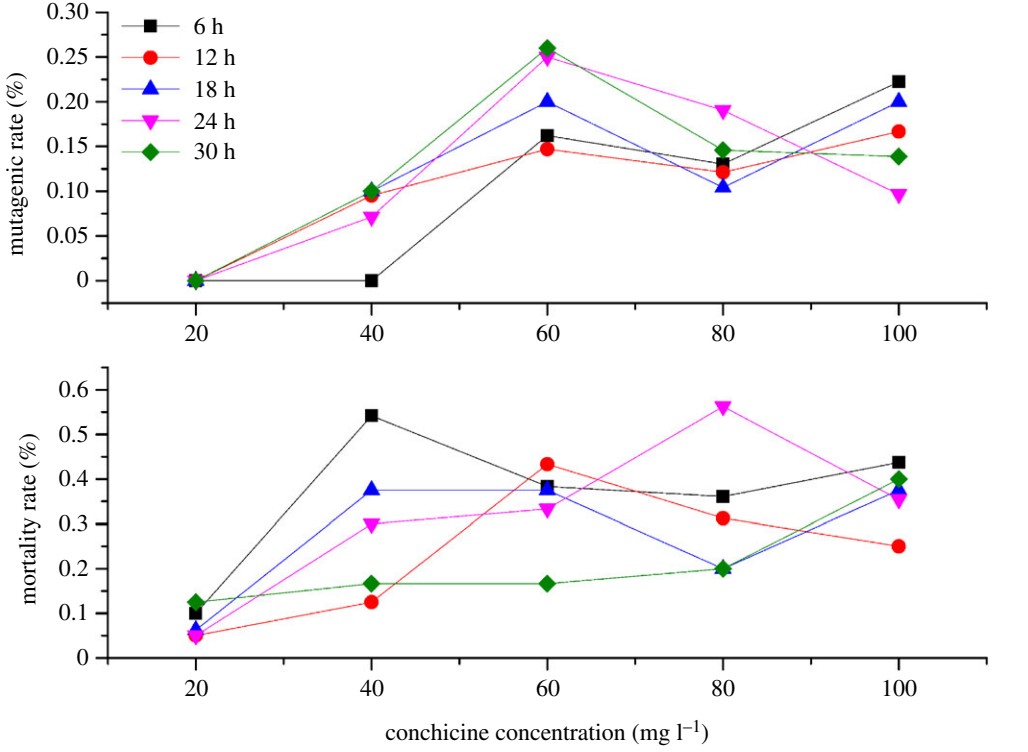

**Figure 1.** The change rule of mutagenic rate and mortality rate.

**Table 4.** Leaf characteristics of diploid and tetraploid plants. Mean separation within column by independent sample *t*-test. The means ($n = 20$) followed by the same letter do not differ at $p = 0.05$.

|  | diploid | tetraploid |
| --- | --- | --- |
| leaf length (cm) | 3.57 | 3.72 |
|  | 0.003 | 0.005 |
|  | b | a |
| leaf width (cm) | 2.26 | 2.35 |
|  | 0.01 | 0.002 |
| leaf thickness (m) | 0.52 | 1.04 |
|  | 0.004 | 0.026 |
|  | B | A |
| stomatal guard cells (μm) | 18 | 12 |
|  | 1.778 | 3.556 |
|  | A | B |

number of stomata on the unit leaf area decreased as the density decreased (figure 2*c*,*d*). The results showed that the average number of the diploid stomatal guard cells within a field of vision was significantly higher than the average number of tetraploid stomatal guard cells ($p = 0.00001795$, table 4).

## 3.3. Chromosome number and nuclear DNA content

Chromosome numbers of diploid *A. chinensis* and tetraploid *A. chinensis* plants were $2n = 2x = 58$ and $2n = 4x = 116$, respectively. Parts of the chromosome number observation results are shown in figure 2*e*,*f*.

As shown in figure 3, the average DNA content of the tetraploid *A. chinensis* plants was about twice that of diploid plants. All 187 suspected tetraploid plants according to morphological indexes were confirmed to be tetraploid.

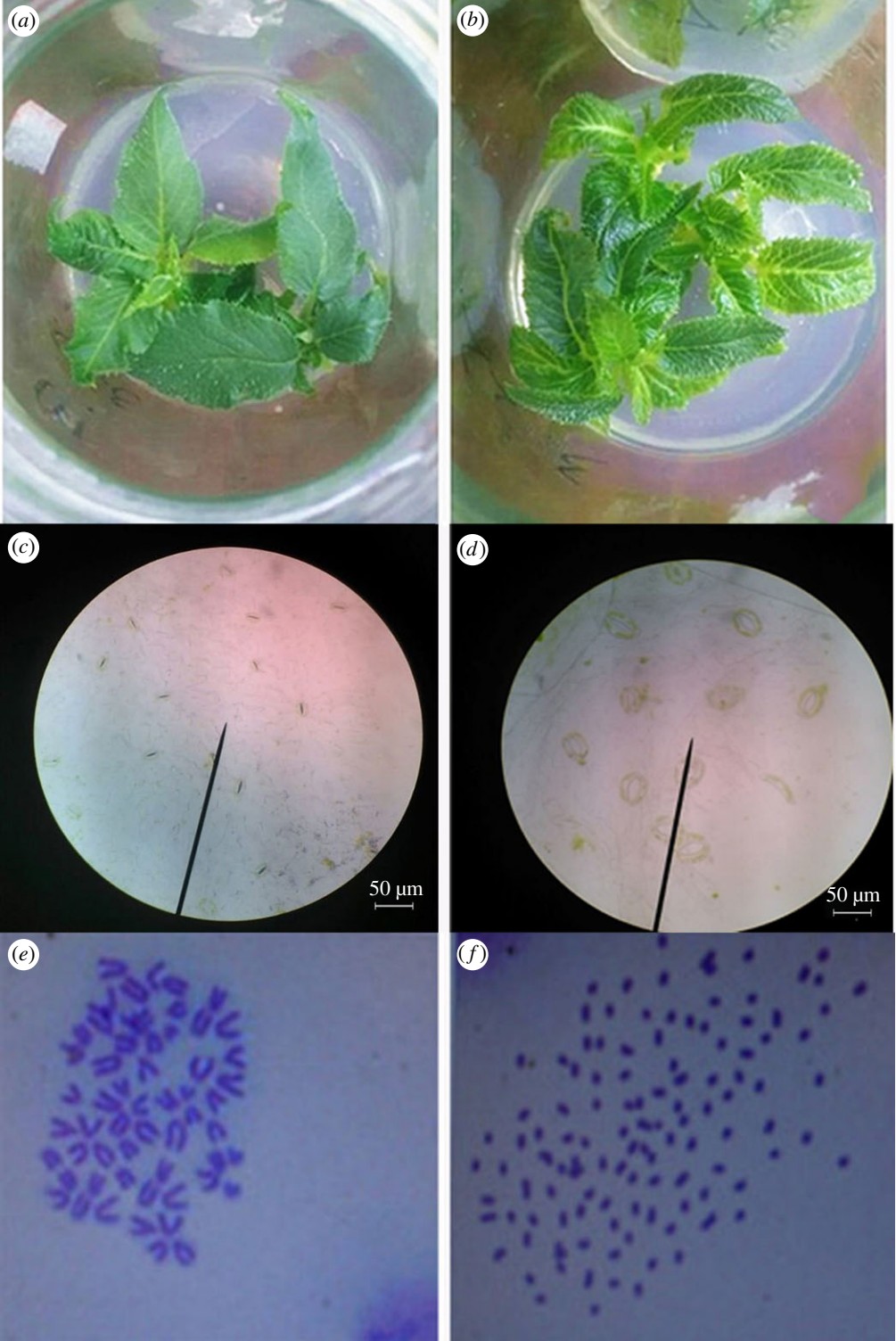

**Figure 2.** The morphological differences, stomatal guard cells and chromosome number of diploid and tetraploid plants. (*a*) diploid plants; (*b*) tetraploid plants with thicker leaves and larger veins; (*c*) stomatal guard cells of diploidy with average length of 15.7 ± 1.44 (μm); (*d*) stomatal guard cells of tetraploidy with average length of 42.1 ± 6.07 (μm); (*e*) chromosome number of diploidy (58); (*f*) chromosome number of tetraploidy (116).

## 3.4. Analysis of differential gene expression and functional annotation

Approximately 47.26 and 41.36 million clean reads were generated for the tetraploid and diploid samples, respectively. Mapping rates of the tetraploid and diploid were 88.01% and 86.53%, respectively, as determined by genome comparison.

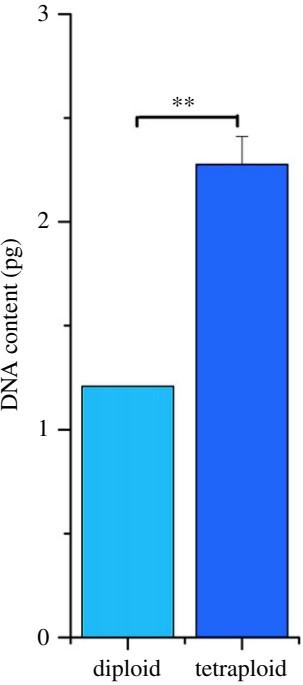

**Figure 3.** Nuclear DNA contents in *A. chinensis* diploid and tetraploid plant leaves determined by flow cytometric analysis. Average DNA content of the diploid *A. chinensis* plants was significantly lower than that of tetraploid plants ($p = 0.000254$). Data were compared using one-way ANOVA and Fisher's post hoc comparisons. $^{**}p < 0.01$.

**Table 5.** DEGs annotation number.

| sample | trend | number of DEGs significantly annotated | the total number of DEGs annotated |
|---|---|---|---|
| diploid versus tetraploid | up | 660 | 16 106 |
| | down | 1570 | 17 943 |
| | total | 2230 | 34 049 |

There involved a total of 34 049 differentially expressed genes between diploid and tetraploid, 16 106 of them were downregulated, and 17 943 of them were upregulated. Among these differentially expressed genes, 2230 were significantly annotated, of which 660 genes downregulated, 1570 genes upregulated in the tetraploid plants opposed to the diploid ones (table 5).

The differentially expressed genes (table 6) were selected from three replicates for which deferentially expressed levels were very significant, and the unannotated genes were removed. Most of the 16 ascending genes were related to growth development and resistance. The gene annotation of mannan synthase [51], vacuolar protein sorting-associated protein [52], auxin efflux carrier protein [53] and cytochrome P450 [54] were related to plant growth and development, and the gene annotation of glycosyltransferase family 2 protein [55] and 3-ketoacyl-CoA synthase [56] were related to plant resistance. In the five descending genes, the three genes, ferritin [57], anthranilate *N*-benzoyltransferase protein [58] and chitinase protein [59] were related to antibacterial activity. The results indicated that the growth and stress resistance of tetraploid plants was better than those of diploid ones, while their antibacterial ability was worse than diploid ones.

## 3.5. Gene ontology classification analysis of DEGs

The differentially expressed genes were subjected to GO (gene ontology) analysis, and the results were reflected in three main categories: cellular components, molecular function and biological process. These DEGs were mainly enriched in GO categories of catalytic activity, binding, transporter activity, membrane and metabolic process.

**Table 6.** DEGs after filtered of tetraploid and diploid A. chinensis.

| gene_id | gene annotation | p-value | regulation | logFC |
|---|---|---|---|---|
| Achn022281 | Patatin T5 | 0.00069357 | up | 2.45684171 |
| Achn072311 | Centrin-1 | $4.88 \times 10^{-9}$ | up | 5.54193126 |
| Achn075911 | putative citrate efflux MATE transporter | 0.000880555 | up | 2.88589185 |
| Achn086711 | mannan synthase | $6.91 \times 10^{-5}$ | up | 4.99262803 |
| Achn107571 | glycosyl transferase family 2 protein | $3.58 \times 10^{-7}$ | up | 5.99001591 |
| Achn133241 | 2-oxoglutarate-dependent dioxygenase | $2.11 \times 10^{-8}$ | up | 4.45289714 |
| Achn150071 | vacuolar protein sorting-associated protein | 0.000797167 | up | 4.41027292 |
| Achn162311 | reductase 1 | 0.000745493 | up | 4.41027292 |
| Achn176611 | cellulose synthase | $1.67 \times 10^{-12}$ | up | 6.62819319 |
| Achn181151 | putative ROX1 | $1.88 \times 10^{-8}$ | up | 5.32241374 |
| Achn207191 | lactoylglutathione lyase | 0.000139736 | up | 2.97970391 |
| Achn211741 | auxin efflux carrier protein | 0.00013422 | up | 4.24269787 |
| Achn263191 | 2-oxoisovalerate dehydrogenase alpha subunit | $2.79 \times 10^{-6}$ | up | 4.46772330 |
| Achn285441 | 1-aminocyclopropane-1-carboxylate oxidase | $4.24 \times 10^{-11}$ | up | 6.02373138 |
| Achn311381 | 3-ketoacyl-CoA synthase | $1.69 \times 10^{-6}$ | up | 3.83116076 |
| Achn329921 | cytochrome P450 | $6.68 \times 10^{-9}$ | up | 5.70327952 |
| Achn036721 | ER glycerol-phosphate acyltransferase | 0.000128046 | up | 2.94934454 |
| Achn103301 | beta-1,3-glucanase | 0.000359838 | down | −2.49679123 |
| Achn175101 | ferritin | $2.48 \times 10^{-8}$ | down | −4.20769625 |
| Achn196611 | anthranilate N-benzoyltransferase protein | 0.000963949 | down | −2.25811355 |
| Achn273491 | chitinase protein | $2.85 \times 10^{-11}$ | down | −5.58243403 |
| Achn334521 | inorganic phosphate transporter | $1.50 \times 10^{-8}$ | down | −4.80873117 |

Using the diploid as the control, the upregulated and downregulated genes in the GO classification were subjected to comparative analysis, and only the genes with altered expression levels were chosen. The results showed that the number of upregulated genes in each GO-term was greater than the downregulated genes (figure 4).

## 3.6. KEGG pathway analysis of DEGs

These DEGs were mainly enriched to Kyoto Encyclopedia of Genes and Genome (KEGG) pathways of fatty acid elongation (ko00062), retinol metabolism (ko00830), linoleic acid metabolism (ko00591), terpenoid backbone biosynthesis (ko00900), sesquiterpenoid and triterpenoid biosynthesis (ko00900), phenylpropanoid biosynthesis (ko00940) and bile secretion (ko04976).

Two complete upward and downward pathways were chosen for analysis (figures 5 and 6). In the pathway of retinol metabolism, there just was upregulated gene, Achn329921. The gene was related to cytochrome P450, which might be highly associated with biosynthesis of terpenoids. The upregulated pathway might be highly related to better growth and development in tetraploid plants than in diploid ones. In the pathway of fatty acid elongation, there were one upregulated gene and two downregulated genes. The upregulated gene Achn311381 was related to 3-ketoacyl-CoA synthase, which was involved in biofilm lipid synthesis, and was a precursor to the formation of waxy organisms in keratoplasm. The two downregulated genes Achn141891 and Achn001331 were related to palmitoyl (protein) and hydrolase.

## 3.7. RT-qPCR validation

To confirm that the differentially expressed unigenes and genes obtained from Illumina sequencing were truly transcribed at different levels between diploid and tetraploid plants, two known genes (figure 7a) associated with stress resistance, three other upregulated genes (figure 7b) and six downregulated genes

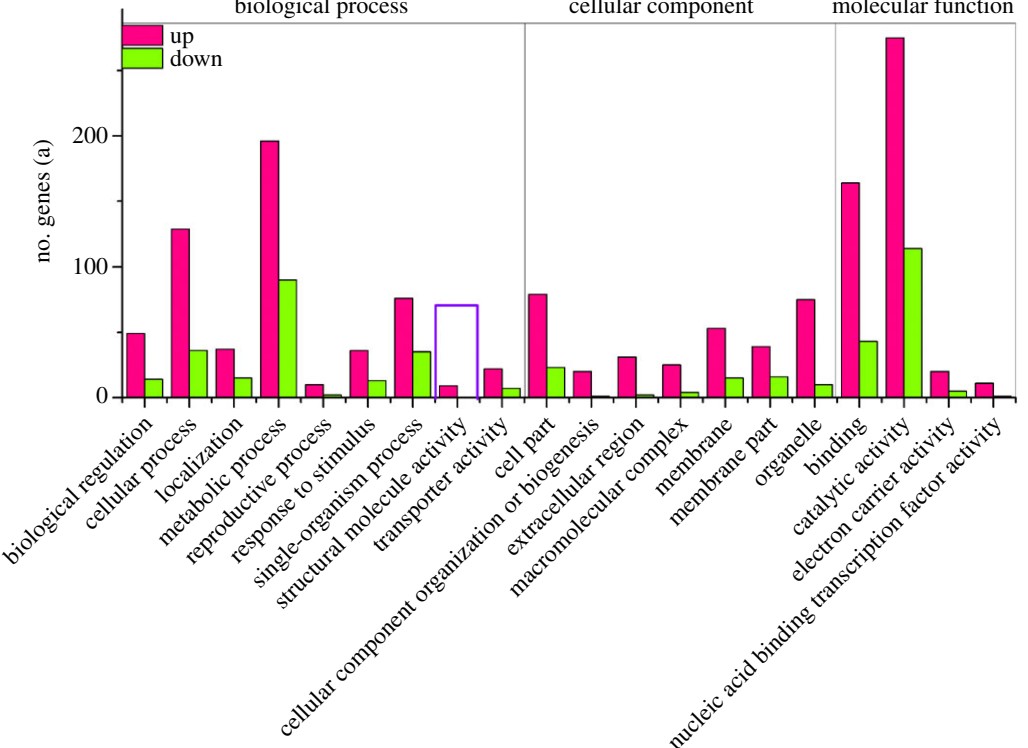

**Figure 4.** Ontology classification of DEGs.

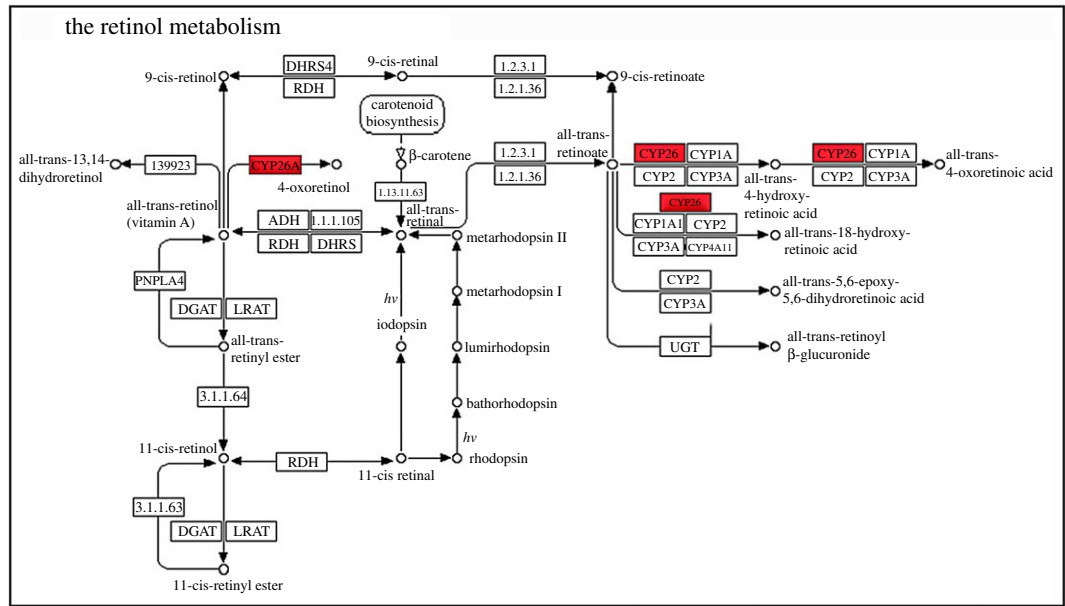

**Figure 5.** The retinol metabolism pathway of A. chinensis diploid and tetraploid plants. Red, upregulated genes; green, downregulated genes; blue, both.

(figure 7c) were chosen for RT-qPCR analysis. The results showed that the three upregulated genes still exhibited enhanced expression in RT-qPCR analysis. Five downregulated genes were confirmed to be consistent with the transcriptome results; however, one gene (Achn007531) was not consistent with it. The reason for this might be the individual differences between initial sequencing and late detection. Overall, the results of RT-qPCR were coherent with the Illumina sequencing. The findings suggested that there were statistically significant differences between the expression levels of these genes in diploid and tetraploid plants.

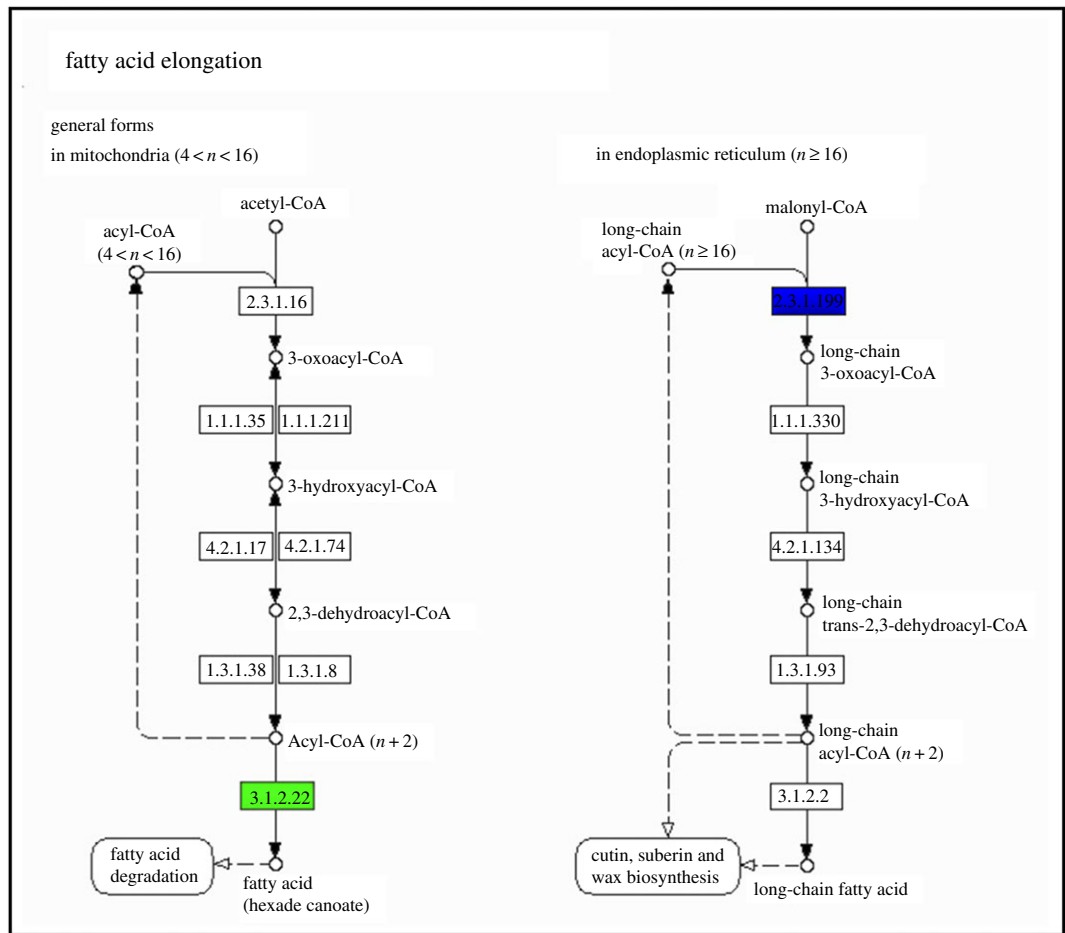

**Figure 6.** The fatty acid elongation pathway of *A. chinensis* diploid and tetraploid plants. Red, upregulated genes; green, downregulated genes; blue, both.

## 3.8. DNA methylation levels of diploid and tetraploid plants

Methylation levels of total amplified sites in diploid and tetraploid tissues were 49.5% and 42.35%, respectively (table 7). Therefore, there was a certain degree of difference among the diploid and tetraploid tissues.

# 4. Discussion

Shoot-immersion method which immersed the shoots with colchicine solution has been used in kiwifruit [60], wheat [61], *Vitis dabidii* [62], maize [63], garlic [64] and other plants to induce polyploidy. The dropping method which dropped colchicine solution on cotton ball binding to the top of shoots was used in *Secale*, *Haynaldia*, *Aegilops*, *Brassica* spp. and mulberry [65,66]. These two methods have their advantages and disadvantages in the induction of tetraploidy. In the study, shoot-immersion method was considered to be better because the tissues of immersion could contact more fully, so the mutation rate would be higher. In practice, the specific conditions of species should be considered in the selection of a suitable method for the induction of tetraploidy.

A previous study showed that using micro shoots treated with 0.05 or 0.1% colchicines effectively induced tetraploidy in *A. chinensis*, and the induction rates were 30% and 20%, respectively [6]. Our results showed that using leaf segments treated with 60 mg l$^{-1}$ (0.006%) also could effectively induce tetraploidy, and the induction rate was about 26%. And we found that using the concentration of 100 mg l$^{-1}$ colchicines (0.01%) the tetraploid induction efficiency was lower than that of 60 mg l$^{-1}$ (0.06%). The concentration of colchicine was 10 times lower than previously reported. It indicated that the different varieties of *A. chinensis* required using different concentrations of colchicine to reach the higher induction rate.

Tetraploid plants show some typical morphological characteristics, like stunted growth and presence of larger, thicker, dark green and larger stomata [67,68]. Morphological difference was also found between

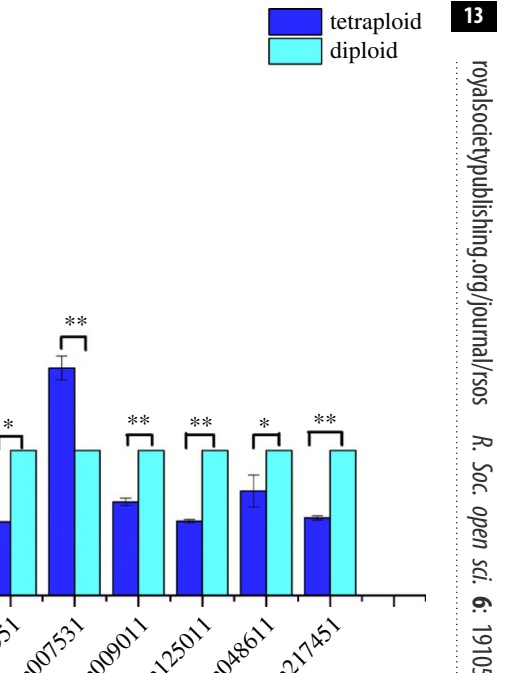

**Figure 7.** The expression levels of the selected genes in diploid and tetraploid plants. (*a*) Two resistance genes; (*b*) three up genes from transcriptome; (*c*) six down genes from transcriptome. Data were compared using one-way ANOVA and Fisher's *post hoc* comparisons. Values were mean ± s.e. of $2^{(-\Delta\Delta Ct)}$ ($n = 3$). *$p < 0.05$; **$p < 0.01$.

**Table 7.** The ratio of patterns of genomic DNA methylation of the diploid and tetraploid tissues. Total methylated bands = II + III + IV; fully methylated bands = III + IV.

| types | diploidy | tetraploid |
|---|---|---|
| I | 366 | 410 |
| II | 22 | 22 |
| III | 216 | 171 |
| IV | 61 | 63 |
| total amplified bands | 604 | 602 |
| total methylated bands | 299 | 255 |
| MSAP (%) | 49.5 | 42.35 |
| fully methylated bands | 277 | 234 |
| fully methylated ratio | 45.86 | 38.87 |

diploid plants and their tetraploid plants in the other research of kiwifruit [8,60]. These external morphologies could be used as evidence for the identification of tetraploid plants. Size and density of stomatal guard cells, chromosome number and flow cytometric analysis were used to identify tetraploid plants in previous study [69,70]. Here, external morphology observation, stomatal guard cell observation, chromosome number observation and flow cytometry analysis were employed to identify the tetraploid of *A. chinensis*. A combination of these methods was found to be effective in the identification of tetraploidy of *A. chinensis*.

There are usually some special characteristics in a tetraploid fruit tree, such as stronger growth with bigger fruit and fewer or no seed, stronger adaption ability and better stress resistance [71]. To our knowledge, there continue to be no reports on investigating the reason for the phenomena at the molecular level. In this study, upregulated genes in DEGs were related to growth and stress resistance. The downregulated genes were related to antibacterial activity. In the GO annotation, all functional genes concerning structural molecule activity were upregulated, indicating that the growth of tetraploid *A. chinensis* plants was stronger than

diploid ones. Meanwhile, almost all the functional genes of membrane and membrane part were upregulated genes, and upregulated expression of the membrane and membrane part-related genes indicated that stress-resistant ability might increase, because the functional genes of biological membrane play very important roles in plant resistance [72]. In the KEGG pathway, the genes associated with stress resistance were upregulated, while those related to leaf senescence were downregulated. Leaf senescence was associated with proteolytic enzymes, and the activity of the proteolytic enzyme increased gradually with leaf senescence [73]. Therefore, opposed to diploid plants, the decreased expression of proteolytic enzyme-related genes could contribute to more robust leaves in tetraploid plants. The transcriptomes would be determined by the full message RNA expression in which the transcription varies depending on many factors; the tetraploids with high transcriptions could be a response to the genomics interaction inside of the new high levels of genomics of the new organize, which could be a natural response [13].

An important development in understanding the influence of chromatin on gene regulation was the finding that DNA methylation and histone post-translational modifications would lead to the recruitment of protein complexes that regulated transcription [74]. DNA methylation might lead to the expression levels of functional genes being increased or decreased. DNA methylation does not alter nucleotide sequences but regulates gene transcription, and it plays an important part in gene expression regulation during development and differentiation in plants [50]. This study adopted methylation-sensitive amplification polymorphism to compare the levels of DNA cytosine methylation in diploid and tetraploid tissues, and the result showed that tetraploid tissues had lower methylation rate than diploid tissues. Here, the increased expression of growth and stress resistance-related genes in the *A. chinensis* tetraploid plants might be caused by DNA methylation too.

The study has, for the first time, found that growth and stress resistance-related genes were upregulated, and antibacterial activity-related genes were downregulated at the transcriptome level in tetraploid plants compared to diploid ones, and tetraploid plants had lower methylation ratio than diploid ones [75,76], which might explain why tetraploid plants usually have better resistance than diploid ones.

## 5. Conclusion

(1) The best treatment for tetraploid induction of *A. chinensis* SWFU 03 was soaking 30 h in 60 mg l$^{-1}$ colchicine solutions. Using the induction system combined with the identification methods including external morphologies comparison and flow cytometry analysis, 187 tetraploid *A. chinensis* plants were obtained in the study.

(2) Compared with diploid plants, the stress tolerance-related genes of tetraploid plants were upregulated. We have, for the first time, found that the growth and stress resistance of tetraploid plants was stronger than those of diploid ones at the transcriptome level.

(3) The tetraploid plants had lower methylation ratio than diploid ones. DNA methylation was important in gene expression regulation during biological development in *A. chinensis* plants.

Data accessibility. RNA-seq data were presented at the short read archive (SRA) database of NCBI (accession number PRJNA535344).

Authors' contributions. S.L. analysed the experimental data and drafted the manuscript. S.L., X.Z. and Q.Y. conducted the study. X.L. helped analyse the experimental data. H.L. participated in the coordination of the study. H.Z. conceived of the study and contributed to writing the manuscript. All authors approved the final version of the manuscript.

Competing interests. There is no any ethical/legal conflicts involved in the article.

Funding. The authors gratefully acknowledge financial support from the Fund for the State Bureau of Forestry 948 Project (2012-4-62) and National Natural Science Foundation of China (grant no. 31360404).

Acknowledgements. The authors thank Research Scientist Xiuyin Chen of the New Zealand Institute for Plant and Food Research for her critical reading of the manuscript.

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
