## [Reviewer comments · Royal Society Open Science]

Review History

RSOS-191052.R0 (Original submission)

Review form: Reviewer 1

Is the manuscript scientifically sound in its present form?

No

Are the interpretations and conclusions justified by the results?

Yes

Is the language acceptable?

No

Do you have any ethical concerns with this paper?

Yes

Have you any concerns about statistical analyses in this paper?

Yes

Recommendation?

Major revision is needed (please make suggestions in comments)

Comments to the Author(s)

The manuscript „Induction, identification and genetics analysis of tetraploid *Actinidia chinensis*” contains very interesting results on comparative analysis of genetic alteration of neotetraploids in relation to their diploid counterpart. Transcriptome analysis revealed significant differences in gene expression between diploid and neotetraploids. However, the work lacks a lot of methodical information. Many sentences are unclear. There are many mental shortcuts and improper terms. English needs significant improvement.

There are the following comments to authors.

Introduction

The introduction should be developed giving some examples for other plant species of the diploid and tetraploid comparative genetic analysis. There are many examples of analyses using RNA-Seq, RT-qPCR and DNA methylation analysis of neotetraploids in relation to their diploid counterpart.

Line 48: give the basic or diploid chromosome number of *A. chinensis* and frequency of tetraploids and other ploidy level polyploids of this species.

Lines 51-52: what are the crossing barriers for diploid and polyploids.

Line 56: the sentence is unclear.

Lines 64-67: the routine, common method for ploidy evaluation/polyploid detection is flow cytometry. Other methods based on the assessment of morphological features are considered auxiliary.

Line 72: “Transcriptome sequencing (RNA-Seq) is based...”. Use the same abbreviation throughout the text, “RNA-seq” is written in several places.

Lines 73-78: please rewrite this part, suggestion: “RNA-Seq is a modern tool of the transcriptome analysis. A new generation of sequencing technology can be used to fast and accurate transcriptome profiling (He and Jiao, 2014). This technology can detect any species of the overall transcription activity; based on the analysis of the transcriptome, gene structure and expression level can be described. This method can also detect novel transcript and rare transcript, provide...”

Line 97: In general, chromosome doubling tends to increase plant resistance to biotic and abiotic stress and can be induced by various antimetabolic agent.” Please add references to this statements.

Line 101: it should be “tetraploids” or “tetraploid”, not “tetraploidy”. Such incorrect term is in many places.

Material and methods

The parameters “mortality rate” and “mutagenic rate” should be changed, explained more carefully and corrected throughout the text, see below.

No information on statistical analysis for phenotypic evaluation and some molecular analyses. In my opinion, flow cytometry analysis and all the molecular methods require more detailed description.

Lines 112-113: it should be: “... an excellent *A. chinensis* breeding clone selected...”

Line 114: it should be: “donor diploid plant was used as reference for phenotypic and genetic analysis of resulted tetraploids.”

Line 119: it should be: “For initiation of in vitro culture,...”

Line 121: “...treatment times...”, not processing times; correct it throughout the text.

Line 122: it should be rather “...each treatment time had a control with...”

Line 125: "Using Murashige and Skoog (1962) medium (MS)...", please add this reference to the list.

Line 126: "(1-naphthylacetic acid) + 5 mg/L BA (benzyladenine)...", the proper abbreviation for benzyladenine is BA and for benzylaminopurine is BAP, although this is the same cytokinin.

Line 129: "Mortality rate", should be rather: "dieback rate".

Line 130: "Mutagenic rate"??, Should be rather "polyploidisation efficiency". It should be explained that this parameter was calculated based on the number of putative tetraploids, selected according to the traits characteristic for tetraploid plants of *A. chinensis*, that was observed in earlier studies.

Please change and explain the parameters "mortality rate" and "mutagenic rate" and correct it throughout the text.

Lines 135-136: the subtitle can be changed to "Phenotypic evaluation"

No information on statistical analysis for phenotypic evaluation.

Line 158: "...relative content of DNA...", should be "relative nuclear DNA content", if it was really measured some more information should be added: what internal standard was used, what kind of instrument for FCM.

Line 163: Insert the abbreviation "...differential expression of genes (DEG)..."

Line 171: it should be "...using DEGseq and DESeq softwares..."

Line 173: it should be tissue culture shoots not seedlings.

Results

In many places it is not known what the results relate to, diploids or tetraploids. Please correct it throughout the text.

Many sentences refer to MM or Discussion and these parts of the text should be transferred to the appropriate chapters.

Subdivision titles should be changed, see below.

Line 192: The subtitle should be changed to, e.g. Polyploidisation efficiency.

Line 193: "fatality rate"???

Line 208: "When treatment time was short brief, the polyploidisation efficiency mutation rate increased gradually...".

Line 216: "... of diploidsy and tetraploidsy...", correct it throughout the text.

Lines 217-218: "the difference was significant", "marginally" should be deleted.

Line 233: The subtitle should be changed to "Chromosome number and nuclear DNA content". There should be information about the number of chromosomes and the value of nuclear DNA content of diploid and tetraploid plants.

Line 237: it should be rather "Analysis of differential gene expression and functional annotation"

Line 260: add full name for GO.

Lines 262-263: "...binding what??, membrane function??" please describe this more carefully.

Lines 266-267: In many places it is not known what the results relate to, diploids or tetraploids. Please correct it throughout the text.

Lines 268-273: This part should be moved to discussion. Please add references.

Line 274: No information on KEGG analysis in MM

Lines 287-288: Do you mean: formation of waxy structures in cuticle??

Lines 289-292: it should be moved to discussion.

Lines 302-303: Information on statistical analysis should be moved to MM.

Line 307: should be "DNA methylation levels of diploid and tetraploid plants"

Lines 308-310: this part should be moved to MM.

Discussion

What is immersion or dropping method, please explain.

The style and grammar of Discussion should be carefully improved.

Tables

Table 1: the 2nd and 4th columns can be removed. Change the title to, e.g. Polyploidisation efficiency using in vitro colchicine treatment of leaf explants of *A. chinensis*. Under the table, add information on the statistical test which was used for mean separation, e.g.: Mean separation within column by any test (give the name). The means (n=20) followed by the same letter do not differ at

$P = 0.05$. Other information is not necessary.

Table 2. The second column should be deleted; "Diploid" and "Tetraploid"; the letters of significance should be given at means as in table 1. Information under the tables as in table 1.

References

Correct alphabetical order of the reference list

Check and correct publication years:

Babil et al. 2011 or 2002

Kobayashi and Shimamura 1954 or 2002

Meaburn et al. 2012 or 2002

The manuscript requires major revision. Then it can be consider for publication.

Decision letter (RSOS-191052.R0)

16-Sep-2019

Dear Dr Zhang,

The editors assigned to your paper ("Induction, identification and genetics analysis of tetraploid *Actinidia chinensis*") have now received comments from reviewers.

The reviewer raises a number of substantive criticisms, including issues surrounding methodological presentation and description. Overall, the reviewer requires major revisions to the manuscript and details a large number of points that will all need addressing carefully.

We would like you to revise your paper in accordance with the referee suggestions which can be found below (not including confidential reports to the Editor). Please note this decision does not guarantee eventual acceptance.

Please submit a copy of your revised paper before 09-Oct-2019. Please note that the revision deadline will expire at 00.00am on this date. If we do not hear from you within this time then it will be assumed that the paper has been withdrawn. In exceptional circumstances, extensions may be possible if agreed with the Editorial Office in advance. We do not allow multiple rounds of revision so we urge you to make every effort to fully address all of the comments at this stage. If deemed necessary by the Editors, your manuscript will be sent back to one or more of the original reviewers for assessment. If the original reviewers are not available, we may invite new reviewers.

To revise your manuscript, log into <http://mc.manuscriptcentral.com/rsos> and enter your

Author Centre, where you will find your manuscript title listed under "Manuscripts with Decisions." Under "Actions," click on "Create a Revision." Your manuscript number has been appended to denote a revision. Revise your manuscript and upload a new version through your Author Centre.

- Data accessibility

If you wish to submit your supporting data or code to Dryad (<http://datadryad.org/>), or modify your current submission to dryad, please use the following link:
<http://datadryad.org/submit?journalID=RSOS&manu=RSOS-191052>

- Competing interests

- Authors' contributions

AB carried out the molecular lab work, participated in data analysis, carried out sequence alignments, participated in the design of the study and drafted the manuscript; CD carried out the statistical analyses; EF collected field data; GH conceived of the study, designed the study,

coordinated the study and helped draft the manuscript. All authors gave final approval for publication.

- Acknowledgements

- Funding statement

on behalf of Dr James Locke (Associate Editor) and Steve Brown (Subject Editor)
openscience@royalsociety.org

Comments to Author:

Reviewers' Comments to Author:

Reviewer: 1

Comments to the Author(s)

The manuscript „Induction, identification and genetics analysis of tetraploid *Actinidia chinensis*” contains very interesting results on comparative analysis of genetic alteration of neotetraploids in relation to their diploid counterpart. Transcriptome analysis revealed significant differences in gene expression between diploid and neotetraploids. However, the work lacks a lot of methodical information. Many sentences are unclear. There are many mental shortcuts and improper terms. English needs significant improvement.

There are the following comments to authors.

Introduction

The introduction should be developed giving some examples for other plant species of the diploid and tetraploid comparative genetic analysis. There are many examples of analyses using RNA-Seq, RT-qPCR and DNA methylation analysis of neotetraploids in relation to their diploid counterpart.

Line 48: give the basic or diploid chromosome number of *A. chinensis* and frequency of tetraploids and other ploidy level polyploids of this species.

Lines 51-52: what are the crossing barriers for diploid and polyploids.

Line 56: the sentence is unclear.

Lines 64-67: the routine, common method for ploidy evaluation/polyploid detection is flow cytometry. Other methods based on the assessment of morphological features are considered auxiliary.

Line 72: "Transcriptome sequencing (RNA-Seq) is based...". Use the same abbreviation throughout the text, "RNA-seq" is written in several places.

Lines 73-78: please rewrite this part, suggestion: "RNA-Seq is a modern tool of the transcriptome analysis. A new generation of sequencing technology can be used to fast and accurate transcriptome profiling (He and Jiao, 2014). This technology can detect any species of the overall transcription activity; based on the analysis of the transcriptome, gene structure and expression level can be described. This method can also detect novel transcript and rare transcript, provide..."

Line 97: In general, chromosome doubling tends to increase plant resistance to biotic and abiotic stress and can be induced by various antimitotic agent." Please add references to this statements.

Line 101: it should be "tetraploids" or "tetraploid", not "tetraploidy". Such incorrect term is in many places.

Material and methods

The parameters "mortality rate" and "mutagenic rate" should be changed, explained more carefully and corrected throughout the text, see below.

No information on statistical analysis for phenotypic evaluation and some molecular analyses. In my opinion, flow cytometry analysis and all the molecular methods require more detailed description.

Lines 112-113: it should be: "... an excellent *A. chinensis* breeding clone selected..."

Line 114: it should be: "donor diploid plant was used as reference for phenotypic and genetic analysis of resulted tetraploids."

Line 119: it should be: "For initiation of in vitro culture,..."

Line 121: "...treatment times...", not processing times; correct it throughout the text.

Line 122: it should be rather "...each treatment time had a control with..."

Line 125: "Using Murashige and Skoog (1962) medium (MS)..." , please add this reference to the list.

Line 126: "(1-naphthylacetic acid) + 5 mg/L BA (benzyladenine)..." , the proper abbreviation for benzyladenine is BA and for benzylaminopurine is BAP, although this is the same cytokinin.

Line 129: "Mortality rate", should be rather: "dieback rate".

Line 130: "Mutagenic rate"??, Should be rather "polyploidisation efficiency". It should be explained that this parameter was calculated based on the number of putative tetraploids, selected according to the traits characteristic for tetraploid plants of *A. chinensis*, that was observed in earlier studies.

Please change and explain the parameters "mortality rate" and "mutagenic rate" and correct it throughout the text.

Lines 135-136: the subtitle can be changed to "Phenotypic evaluation"

No information on statistical analysis for phenotypic evaluation.

Line 158: "...relative content of DNA..." , should be "relative nuclear DNA content", if it was really measured some more information should be added: what internal standard was used, what kind of instrument for FCM.

Line 163: Insert the abbreviation "...differential expression of genes (DEG)..."

Line 171: it should be "...using DEGseq and DESeq softwares..."

Line 173: it should be tissue culture shoots not seedlings.

Results

In many places it is not known what the results relate to, diploids or tetraploids. Please correct it throughout the text.

Many sentences refer to MM or Discussion and these parts of the text should be transferred to the appropriate chapters.

Subdivision titles should be changed, see below.

Line 192: The subtitle should be changed to, e.g. Polyploidisation efficiency.

Line 193: "fatality rate"???

Line 208: "When treatment time was short brief, the polyploidisation efficiency mutation rate increased gradually...".

Line 216: "... of diploidsy and tetraploidsy...", correct it throughout the text.

Lines 217-218: "the difference was significant", "marginally" should be deleted .

Line 233: The subtitle should be changed to "Chromosome number and nuclear DNA content". There should be information about the number of chromosomes and the value of nuclear DNA content of diploid and tetraploid plants.

Line 237: it should be rather "Analysis of differential gene expression and functional annotation"
Line 260: add full name for GO.

Lines 262-263: "...binding what??, membrane function??" please describe this more carefully.

Lines 266-267: In many places it is not known what the results relate to, diploids or tetraploids. Please correct it throughout the text.

Lines 268-273: This part should be moved to discussion. Please add references.

Line 274: No information on KEGG analysis in MM

Lines 287-288: Do you mean: formation of waxy structures in cuticle??

Lines 289-292: it should be moved to discussion.

Lines 302-303: Information on statistical analysis should be moved to MM.

Line 307: should be " DNA methylation levels of diploid and tetraploid plants

Lines 308-310: this part should be moved to MM.

Discussion

What is immersion or dropping method, please explain.

The style and grammar of Discussion should be carefully improved.

Tables

Table 1: the 2nd and 4th columns can be removed. Change the title to, e.g. Polyploidisation efficiency using in vitro colchicine treatment of leaf explants of *A. chinensis* . Under the table, add information on the statistical test which was used for mean separation, e.g.: Mean separation within column by any test (give the name). The means (n=20) followed by the same letter do not differ at

$P = 0.05$. Other information is not necessary.

Table 2. The second column should be deleted; "Diploid" and "Tetraploid"; the letters of significance should be given at means as in table 1. Information under the tables as in table 1.

References

Correct alphabetical order of the reference list

Check and correct publication years:

Babil et al. 2011 or 2002

Kobayashi and Shimamura 1954 or 2002

Meaburn et al. 2012 or 2002

The manuscript requires major revision. Then it can be consider for publication.

Author's Response to Decision Letter for (RSOS-191052.R0)

See Appendix A.

Decision letter (RSOS-191052.R1)

16-Oct-2019

Dear Dr Zhang,

I am pleased to inform you that your manuscript entitled "Induction, identification and genetics analysis of tetraploid *Actinidia chinensis*" is now accepted for publication in Royal Society Open Science.

In preparing the final manuscript for publication, please take note of the sentence highlighted by the Associate Editor which needs attention and correction.

Kind regards,
Anita Kristiansen
Editorial Coordinator
Royal Society Open Science
openscience@royalsociety.org

on behalf of Dr James Locke (Associate Editor) and Steve Brown (Subject Editor)
openscience@royalsociety.org

Associate Editor Comments to Author (Dr James Locke):
Associate Editor

Comments to the Author:

Thank you for addressing the reviewer comments. The work is acceptable for publication, although the English remains a little hard to follow. I was unsure what the sentence 'Researchers

had attempted on *A. chinensis* with ploidy level manipulation (Wang et al., 2006; Wu et al., 2009; Wu et al., 2012).¹ meant for example. Do you mean

¹Researchers had attempted ploidy level manipulation on *A. chinensis* (Wang et al., 2006; Wu et al., 2009; Wu et al., 2012).

If possible, it would be great if you could carefully check the manuscript for grammatical errors.

Appendix A

Dear editors :

On behalf of my co-authors, we thank you very much for giving us an opportunity to revise our manuscript, we appreciate editor and reviewers very much for their positive and constructive comments and suggestions on our manuscript entitled “Induction, identification and genetics analysis of tetraploid *Actinidia chinensis*”. We have studied reviewer’s comments carefully and have made revision which marked in red in the paper. We have tried our best to revise our manuscript according to the comments. Attached please find the revised version, which we would like to submit for your kind consideration. We would like to express our great appreciation to you and reviewers for comments on our paper. Looking forward to hearing from you. Thank you and best regards.

Yours sincerely,

Corresponding author: Zhang Hanyao

Response to reviewer:

Thank you for your letter and for the reviewers’ comments concerning our manuscript entitled “Induction, identification and genetics analysis of tetraploid *Actinidia chinensis*”. Those comments are all valuable and very helpful for revising and improving our paper, as well as the important guiding significance to our researches. We have studied comments carefully and have made correction which we hope meet with approval. **Revised portion are marked in red in the paper.** The main corrections in the paper and the responds to the reviewer’s comments are as flowing:

Reviewer:

1. **In addition to addressing all of the reviewers' and editor's comments please also ensure that your revised manuscript contains the following sections as appropriate before the reference list.**

Response to comment: Ethics statement, Data accessibility

Competing interests, Authors’ contributions, Acknowledgements and Funding statement had been added in the format of the journal.

2. **Introduction**

line 48: give the basic or diploid chromosome numer of *A. chinensis* and frequency of tetraploids and other ploidy level polyploids of this species.

Lines 51-52: what are the crossing barriers for diploid and polyploids.

Line 56: the sentence is unclear.

Response to comment: The above items have been added and rewritten based on comments.

Lines 64-67: the routine, common method for ploidy evaluation/polyploid detection is flow cytometry. Other methods based on the assessment of morphological features are considered auxiliary.

Line 72: “Transcriptome sequencing (RNA-Seq) is based...”. Use the same abbreviation throughout the text, "RNA-seq" is written in several places.

Lines 73-78: please rewrite this part, suggestion: “RNA-Seq is a modern tool of the transcriptome analysis. A new generation of sequencing technology can be used to fast and accurate transcriptome profiling (He and Jiao, 2014). This technology can detect any species of the overall transcription activity; based on the analysis of the transcriptome, gene structure and expression level can be described. This method can also detect novel transcript and rare transcript, provide...”

Line 97: In general, chromosome doubling tends to increase plant resistance to biotic and abiotic stress and can be induced by various antimetabolic agent.” Please add references to this statements.

Line 101: it should be “tetraploids” or “tetraploid”, not “tetraploidy”. Such incorrect term is in many places.

Response to comment: The above items have been revised according to the revised comments.

3. Material and methods

Lines 112-113: it should be: “... an excellent *A. chinensis* breeding clone selected...”

Line 114: it should be: “donor diploid plant was used as reference for phenotypic and genetic analysis of resulted tetraploids.”

Line 119: it should be: “For initiation of in vitro culture,...”

Line 121: “...treatment times...”, not processing times; correct it throughout the text.

Line 122: it should be rather “...each treatment time had a control with...”

Line 125: “Using Murashige and Skoog (1962) medium (MS)...”, please add this reference to the list.

Line 126: “(1-naphthylacetic acid) + 5 mg/L BA (benzyladenine)...”, the proper abbreviation for benzyladenine is BA and for benzylaminopurine is BAP, although this is the same cytokinin.

Line 129: “Mortality rate”, should be rather : “dieback rate”.

Line 130: “Mutagenic rate”??, Should be rather “polyploidisation efficiency”. It should be explained that this parameter was calculated based on the number of putative tetraploids, selected according to the traits characteristic for tetraploid plants of *A. chinensis*, that was observed in earlier studies.

Please change and explain the parameters “mortality rate” and “mutagenic rate” and correct it

throughout the text.

Line 163: Insert the abbreviation "...differential expression of genes (DEG)..."

Line 171: it should be "...using DEGseq and DESeq softwares..."

Line 173: it should be tissue culture shoots not seedlings.

Response to comment: The above items have been revised according to the revised comments.

Lines 135-136: the subtitle can be changed to "Phenotypic evaluation"

No information on statistical analysis for phenotypic evaluation.

Line 158: "...relative content of DNA...", should be "relative nuclear DNA content", if it was really measured some more information should be added: what internal standard was used, what kind of instrument for FCM.

Response to comment: The above items have been added and rewritten based on comments.

4. Results

Line 192: The subtitle should be changed to, e.g. Polyploidisation efficiency.

Line 193: "fatality rate"??? dieback rate

Line 208: "When treatment time was short brief, the polyploidisation efficiency mutation rate increased gradually..."

Line 216: "... of diploidsy and tetraploidsy...", correct it throughout the text.

Lines 217-218: "the difference was significant", "marginally" should be deleted .

Line 233: The subtitle should be changed to "Chromosome number and nuclear DNA content".

There should be information about the number of chromosomes and the value of nuclear DNA content of diploid and tetraploid plants.

Line 237: it should be rather "Analysis of differential gene expression and functional annotation"

Line 260: add full name for GO.

Lines 262-263: "...binding what??. membrane function??" please describe this more carefully.

Lines 266-267: In many places it is not known what the results relate to, diploids or tetraploids. Please correct it throughout the text.

Lines 268-273: This part should be moved to discussion. Please add references.

Line 274: No information on KEGG analysis in MM

Lines 287-288: Do you mean: formation of waxy structures in cuticle?? yes

Lines 289-292: it should be moved to discussion.

Lines 302-303: Information on statistical analysis should be moved to MM.

Line 307: should be "DNA methylation levels of diploid and tetraploid plants

Lines 308-310: this part should be moved to MM.

Response to comment: The above items have been revised according to the revised comments.

5. Discussion

What is immersion or dropping method, please explain.

The style and grammar of Discussion should be carefully improved.

Response to comment: The above items have been added and rewritten based on comments.

6. Tables

Table 1: the 2nd and 4th columns can be removed. Change the title to, e.g. Polyploidisation efficiency using in vitro colchicine treatment of leaf explants of *A. chinensis*. Under the table, add information on the statistical test which was used for mean separation, e.g.: Mean separation within column by any test (give the name). The means (n=20) followed by the same letter do not differ at $P = 0.05$. Other information is not necessary.

Table 2. The second column should be deleted; “Diploid” and “Tetraploid”; the letters of significance should be given at means as in table 1. Information under the tables as in table 1.

Response to comment: The above items have been revised according to the revised comments.

7. References

Correct alphabetical order of the reference list

Check and correct publication years

Response to comment: The above items have been revised according to the revised comments.